# Intensive care unit sinks are persistently colonized with multidrug resistant bacteria and mobilizable, resistance-conferring plasmids

Luke Diorio-Toth,[1] Meghan A. Wallace,[2] Christopher W. Farnsworth,[2] Bin Wang,[1,2] Danish Gul,[3] Jennie H. Kwon,[4] Saadia Andleeb,[3] Carey-Ann D. Burnham,[2,4,5,6] Gautam Dantas[1,2,5,6,7]

**ABSTRACT**   Contamination of hospital sinks with microbial pathogens presents a serious potential threat to patients, but our understanding of sink colonization dynamics is largely based on infection outbreaks. Here, we investigate the colonization patterns of multidrug-resistant organisms (MDROs) in intensive care unit sinks and water from two hospitals in the USA and Pakistan collected over 27 months of prospective sampling. Using culture-based methods, we recovered 822 bacterial isolates representing 104 unique species and genomospecies. Genomic analyses revealed long-term colonization by *Pseudomonas* spp. and *Serratia marcescens* strains across multiple rooms. Nanopore sequencing uncovered examples of long-term persistence of resistance-conferring plasmids in unrelated hosts. These data indicate that antibiotic resistance (AR) in *Pseudomonas* spp. is maintained both by strain colonization and horizontal gene transfer (HGT), while HGT maintains AR within *Acinetobacter* spp. and Enterobacterales, independent of colonization. These results emphasize the importance of proactive, genomic-focused surveillance of built environments to mitigate MDRO spread.

**IMPORTANCE**   Hospital sinks are frequently linked to outbreaks of antibiotic-resistant bacteria. Here, we used whole-genome sequencing to track the long-term colonization patterns in intensive care unit (ICU) sinks and water from two hospitals in the USA and Pakistan collected over 27 months of prospective sampling. We analyzed 822 bacterial genomes, representing over 100 different species. We identified long-term contamination by opportunistic pathogens, as well as transient appearance of other common pathogens. We found that bacteria recovered from the ICU had more antibiotic resistance genes (ARGs) in their genomes compared to matched community spaces. We also found that many of these ARGs are harbored on mobilizable plasmids, which were found shared in the genomes of unrelated bacteria. Overall, this study provides an in-depth view of contamination patterns for common nosocomial pathogens and identifies specific targets for surveillance.

**KEYWORDS**   whole-genome sequencing, antimicrobial resistance, horizontal gene transfer, plasmid ecology, hospital surveillance, genomic epidemiology

The spread of antibiotic resistance (AR) poses a global threat to the healthcare system, increasing morbidity and mortality associated with infectious diseases (1–3). Globally, there were an estimated 4.95 million deaths associated with bacterial AR in 2019, and deaths attributable to AR in European Union have increased ~2.5-fold between 2007 and 2015 (1, 4). Hospitalized patients are especially vulnerable to infections by multidrug-resistant organisms (MDROs) (5–9). One model of how hospital-acquired infections (HAIs) occur is that shedding of pathogens by a patient or

Address correspondence to Saadia Andleeb, saadiamarwat@yahoo.com, Carey-Ann D. Burnham, cburnham@wustl.edu, or Gautam Dantas, dantas@wustl.edu.

The authors declare no conflict of interest.

See the funding table on p. 18.

healthcare worker seeds contamination of common surfaces and equipment, which can in turn seed infection of additional patients (5, 10–12). In fact, contaminated hospital surfaces have been clearly linked to specific outbreaks of infections caused by lineages of related MDROs, which colonize plumbing systems and spread on hospital surfaces (13–17). However, there are still key gaps in our understanding of the natural history of MDRO colonization dynamics in the hospital built environment. First, much of the prior work in this space has focused on retrospective sampling in the context of outbreaks, which provides an important yet incomplete picture of MDRO colonization dynamics in healthcare systems. Furthermore, although genomic surveillance of nosocomial pathogens is common, the choice of methodology has a large impact on discriminatory index. Specifically, methods that can differentiate between isolates that belong to a common endemic lineage versus those that result from a recent transmission are not commonly used, resulting in a course-grained picture of transmission dynamics (18). To identify recent transmission events, single-nucleotide polymorphisms (SNPs) tracking is required to discriminate between lineages that may be endemic to a region (19). Additionally, long-term contamination can lead to MDROs transferring genes conferring AR, heightened virulence, and environmental persistence to other species via horizontal gene transfer (HGT) (20). Transmission of mobile genetic elements (MGEs) has previously been shown to exacerbate nosocomial outbreaks (21–24) and mediate multidrug resistance across large phylogenetic distances (25–28). A higher-resolution understanding of the composition and genomic adaptations of the hospital built environment microbiome, and scope of HGT that occurs is needed to reduce the burden of HAIs.

Previously, we found that healthcare surfaces at a tertiary care hospital in Pakistan (PAK-ICU) carried a high burden of MDROs that were dominated by closely related lineages, and limited long-read sequencing suggested widespread HGT of the carbapenemase $bla_{NDM-1}$ (29). Although we recovered many common pathogens that could be attributed to shedding from fecal contamination, we also observed extensive contamination by opportunistic, soil- and water-associated pathogens, such as *Pseudomonas stutzeri*. This led us to hypothesize that water sources—which are difficult to decontaminate due to the formation of biofilms—were acting as a significant reservoir for these environmental MDROs (30, 31). To study the patterns and persistence of MDROs and antibiotic resistance genes (ARGs) in hospital water systems, we conducted a new 5-month-long longitudinal study at PAK-ICU, where endemic burden of ARGs is high (Fig. 1A) (1, 29, 32, 33). To determine if the hospital environment represented a different MDRO burden compared to the local environment, we also sampled two community environments: two private homes (PAK-HOME) and two office break rooms (PAK-WORK). As different ARGs have been found in genetically similar MDR Enterobacterales isolated from hospitals in Pakistan and the USA (32), we sampled matched sites in the USA (US-ICU, US-HOME, and US-WORK). Each sample was subjected to a suite of selective culturing designed to enrich for MDROs (see Materials and Methods), and isolates were characterized by whole-genome sequencing (WGS).

From our 28 months of sampling over these two studies, we show a high burden of common nosocomial pathogens on sink surfaces—including many Pseudomonadales and Enterobacterales—and strains that persist throughout PAK-ICU for as long as 2 years. We found that isolates recovered from ICUs have higher ARG abundance and diversity compared to those collected at HOME and WORK sites, and ARGs that confer resistance to antibiotics of last-resort are found in both common and opportunistic pathogens. Finally, we show cross-species sharing of plasmids that confer clinical resistance to all beta-lactams tested, including carbapenems. Altogether, this presents a concerning scenario where the PAK-ICU water system allows for persistence of MDROs through vertical transmission of related clones and transmission of resistance-conferring MGEs between taxa by HGT. These results demonstrate the importance of methodically characterizing hospital microbiomes in a surface-focused manner, to better understand how MDROs move through and persist in the healthcare environment.

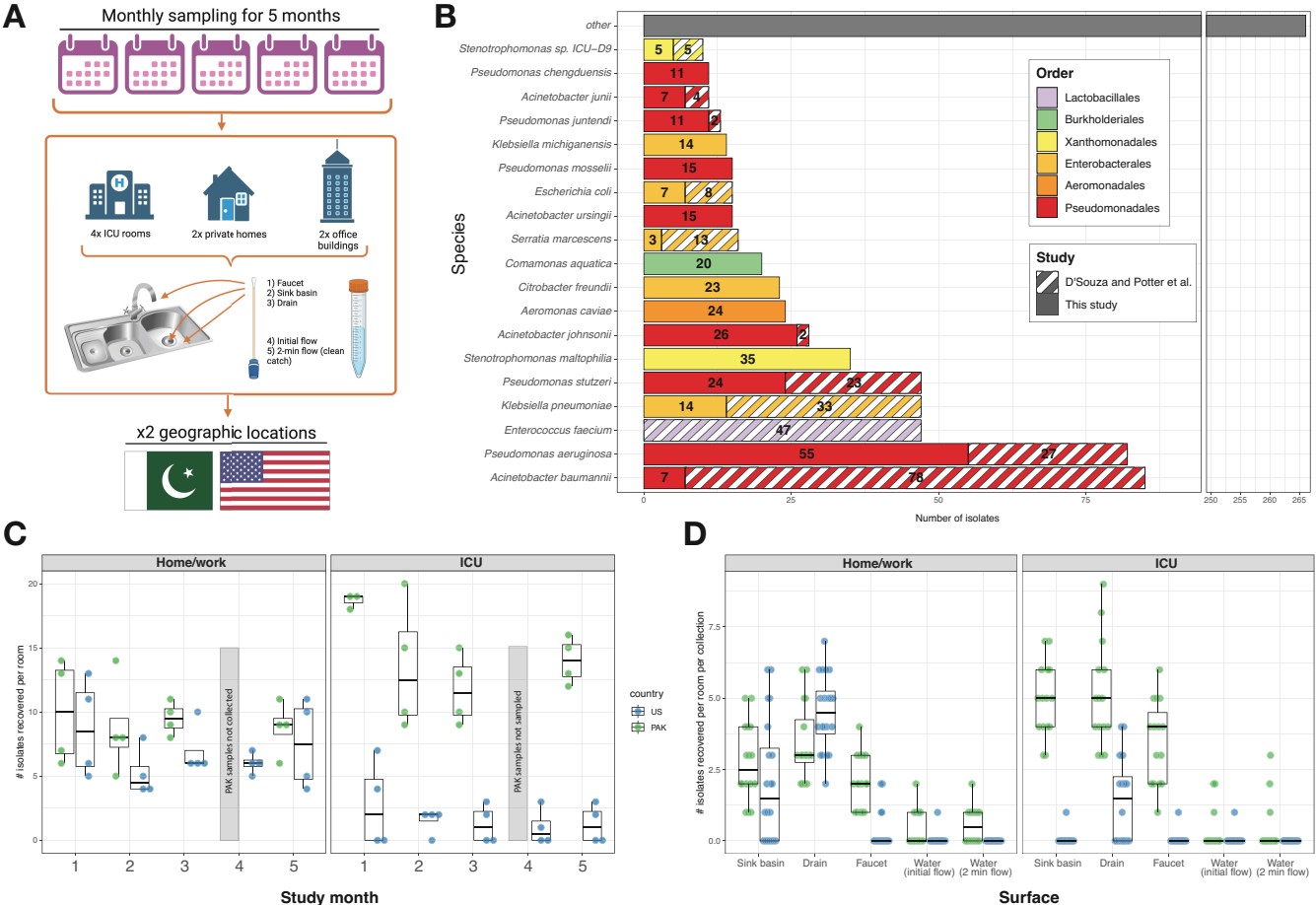

**FIG 1** Study overview. (A) Overview of collection scheme. Created with BioRender.com. (B) Most commonly recovered species from both this study and D'Souza et al., colored by order. (C) Number of isolates recovered per room, per time point. Faceted by environment and colored by country. Due to a local holiday, Month 4 samples were not collected from PAK sites. (D) Number of isolates collected per room, per collection. Faceted by environment and colored by country.

## RESULTS

### PAK and US sinks and water have a high burden of species of *Pseudomonas*, *Acinetobacter*, and *Stenotrophomonas*

In this study, we recovered 530 bacterial isolates from 360 samples taken from hospital, office, and home water and sink surfaces in Pakistan and the United States. To improve taxonomic resolution, resolve transmission dynamics, and analyze ARG content, we performed WGS on all isolates. In total, we analyzed 822 recovered isolates, including the 292 isolates from PAK-ICU and US-ICU previously sequenced by D'Souza et al. (29). These isolates represented 67 different species and 37 genomospecies (Fig. 1B, Supplementary Data), revealing that our current water-focused survey sampled a distinct yet overlapping microbial ecosystem compared to our previous study. In contrast to D'Souza et al., where 28.0% (81/292) of the isolates were either *Acinetobacter baumannii* or *Enterococcus faecium*, in this study we recovered few *A. baumannii* (5) and no *E. faecium*. The most common species was *Pseudomonas aeruginosa*, which represented 10.4% (55/530) of all isolates in this study. Other common pathogens, such as *Klebsiella pneumoniae* (14) and *Escherichia coli* (5), were recovered less frequently. The majority of isolates were identified as common soil- and water-associated opportunistic pathogens, such as *Stenotrophomonas maltophilia* (34), *Acinetobacter johnsonii* (26), *Aeromonas caviae* (24), *P. stutzeri* (24), and *Citrobacter freundii* (23). Overall, 34% (181/530) of the isolates were identified as *Pseudomonas*, 12.1% (64/530) were *Acinetobacter*, and 11.9% (63/530)

were *Stenotrophomonas* (Fig. S1A). Within our collection, some taxa were exclusive to a country: *P. stutzeri* was recovered only from PAK sites, and *Acinetobacter ursingii* was recovered only from US sites (Fig. S1C). Like D'Souza et al., more isolates were recovered from PAK-ICU compared to US-ICU (213 vs 31, respectively). The difference in microbial recovery for HOME and WORK sites between countries was smaller ($n$ = 148 in PAK vs $n$ = 138 in USA) (Fig. 1C). Across all environments, sink surfaces (sink basin, drain trap, and faucet) yielded the most isolates per collection compared to the water itself (Fig. 1D). In US-ICU, nearly all isolates (28/31) were recovered from drains (Fig. 1D). Despite the diversity of individual species recovered, the most common taxa were collected at multiple time points, suggesting MDRO strains may persist in the sampled environments.

## *P. aeruginosa*, *P. stutzeri*, and *Serratia marcescens* strains persist on PAK-ICU surfaces for more than 2 years

We hypothesized that PAK-ICU sink surfaces are persistently colonized by closely related isolates. To identify possible transmission events, we first constructed a core genome alignment of the draft genomes for the most common species and grouped genomes into lineages of <500 core genome SNPs. We next aligned reads to assemblies within each group in an all-vs-all manner to select an internal reference and counted whole genome single-nucleotide variants (SNVs) against that reference. To define strain-level groups that were most likely the result of a recent transmission, we used a cutoff of ≥99.9995% average nucleotide identity (ANI) based on SNVs to an in-group reference (see Materials and Methods) (18). We found that many of the frequently recovered isolates were members of closely related strains, some appearing transiently and other persisting for at least 2 years (Fig. 2). *P. aeruginosa* genomes were commonly found in strain groups (<36 SNVs), with 86.6% (71/82) belonging to 1 of 13 strains (Fig. 2; Fig. 3A and B). Similarly, 85.7% (30/35) of *S. maltophilia* genomes belonged to one of three strains (<23 SNVs), with a single strain dominating PAK samples (Fig. S2A and B). Individual isolates of *S. maltophilia* Strain 1 were recovered from four different buildings in Pakistan (PAK-WORK and PAK-HOME), both on sink surfaces and in the water itself (Fig. S2A and B). Interestingly, no *S. maltophilia* isolates were recovered from PAK-ICU. Many *P. aeruginosa* strains were time- and space-restricted, such as Strains 10, 12, and 13, which were recovered from a single room at a single time point (Fig. 3A and B). However, most strains were recovered over longer periods of time, such as Strain 2, where the time between the first and last appearance was 27 months (Fig. 3B). Including isolates reported by D'Souza et al. (which did not sample exclusively from water-associated surfaces), we found Strain 2 isolates on the sink handles, alcohol foam dispenser, and nurse call button, highlighting that contamination is not limited to sinks (29). Strain 2 isolates were recovered from multiple different rooms in PAK-ICU, suggesting a common, local source of this strain in this environment. This pattern of *P. aeruginosa* colonization was not limited to PAK-ICU, as Strain 3 isolates were detected in three different US-ICU sinks over a 4-month period (Fig. 3B).

*P. stutzeri* showed similar persistence patterns as *P. aeruginosa*, where some strains were transient, while one (Strain 3) was detected over a period of about 26 months in multiple rooms (Fig. S2E and F). *Serratia marcescens* was also persistent over a period of about 24 months, but dominated by a single strain that was detected in seven PAK-ICU rooms (Fig. S2C and D). Although not as extensive in time scale, Enterobacterales such as *Klebsiella*, *Escherichia*, and *Citrobacter* were found in strain groups at multiple times during the study (Fig. S2G through M). By contrast, *A. johnsonii* and *A. junii* strains were exclusively found in the same room at the same time point, despite being commonly found on sink surfaces (Fig. 3C through E). This suggests that strain persistence is either not as common or persisting strains exist at a lower density compared to other taxa.

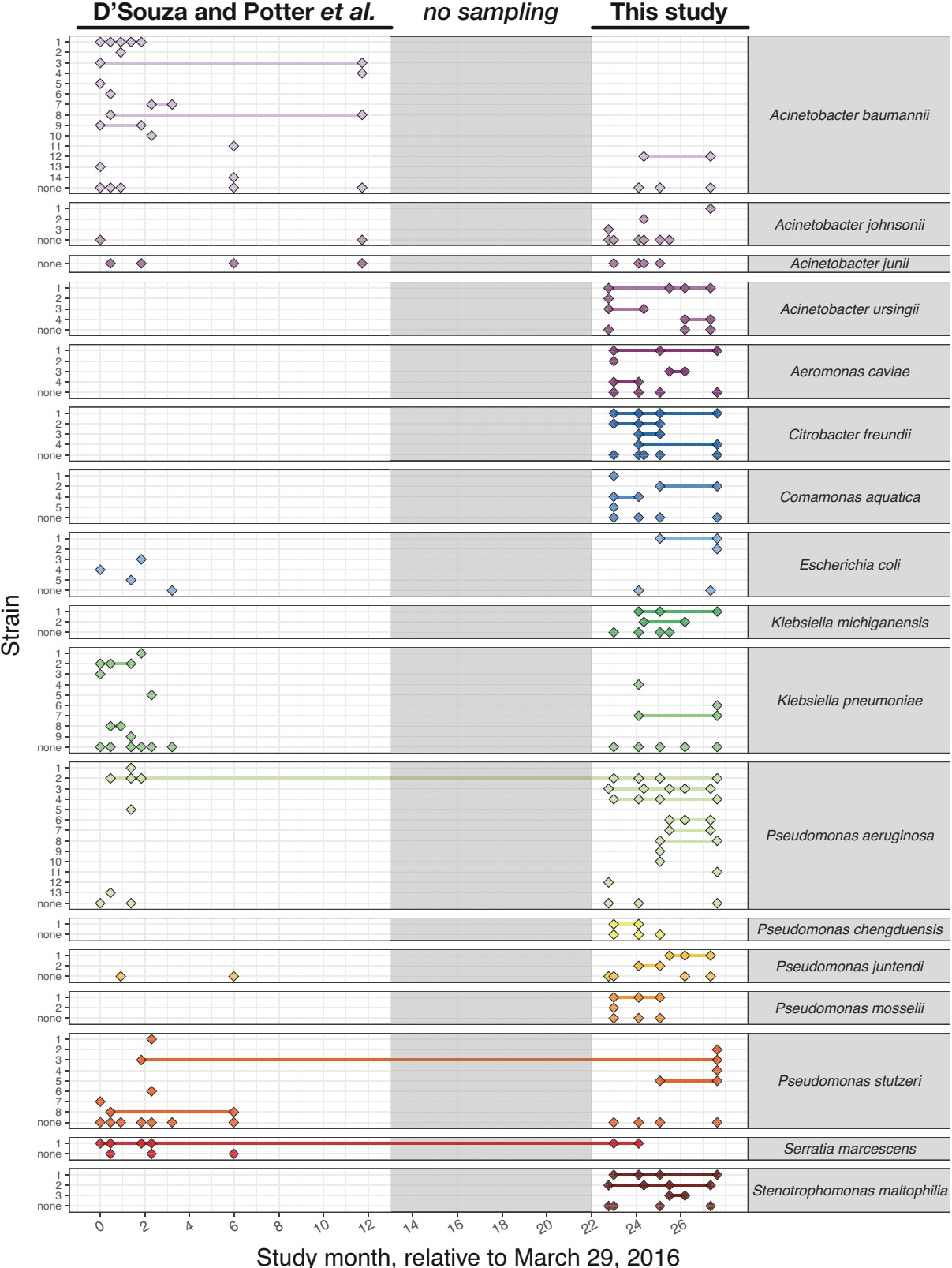

**FIG 2** Overview of strains identified in commonly recovered isolates. Rows represent single strains, and diamonds represent a time point where that strain was identified. Lines drawn between collections of the same strain to highlight persistence. The time between both studies, where samples were not collected, is grayed out.

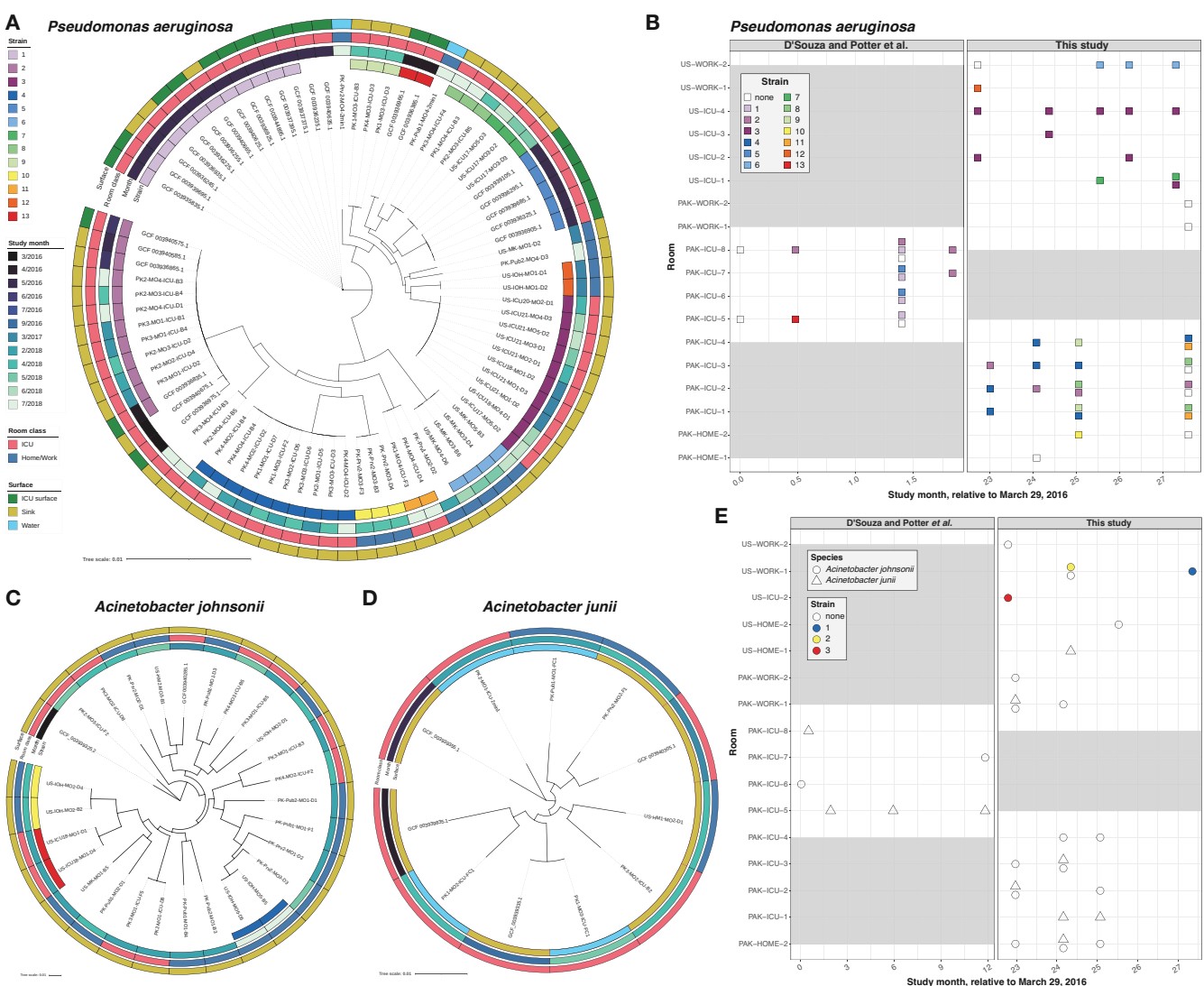

**FIG 3** (A) Maximum likelihood phylogenetic tree of *Pseudomonas aeruginosa* isolates, generated from core genome alignment. Metadata are colored as rings (from inside out): Strain, Study month, Room class, and Surface. (B) Space and time information for *P. aeruginosa* isolates. Squares represent at least one isolate was recovered at that time point, and rows represent different rooms. Squares colored by strain identity. Gray blocks indicate no sampling was performed at those time points. (C) Maximum likelihood phylogenetic tree of *Acinetobacter johnsonii* isolates. (D) Maximum likelihood phylogenetic tree of *A. junii* isolates. (E) Space and time information for *Acinetobacter* isolates. Shape represents species, and as in panel (C), the *x*-axis represents study month and the *y*-axis represents room. Gray blocks indicate no sampling was performed at those time points.

## Isolates recovered from ICU rooms were enriched in ARG abundance and diversity compared to HOME and WORK rooms

Isolates recovered from ICUs had a higher abundance of ARGs compared to isolates from HOME or WORK rooms, in both PAK and US sites ($P < 2.22e-16$ and $P = 4.4e-07$, Wilcoxon rank-sum test) (Fig. 4A). We also found that isolates recovered from ICUs possessed ARGs that were predicted to have activity against more drug classes, in both PAK and US sites ($P < 2.22e-16$ and $P = 2.5e-05$, Wilcoxon rank-sum test) (Fig. 4B). As taxa were differentially recovered from different environments, we compared the ARG abundance within each genus. Genera that were exclusively recovered from ICUs, such as *Escherichia* and *Serratia,* had a higher burden of ARGs compared to genera that were exclusively recovered from HOME or WORK rooms, such as *Agrobacterium* (Fig. 4C and D). But even within genera recovered from both environments, isolates that were recovered from ICUs had significantly higher abundance and diversity of ARGs ($P < 0.05$,

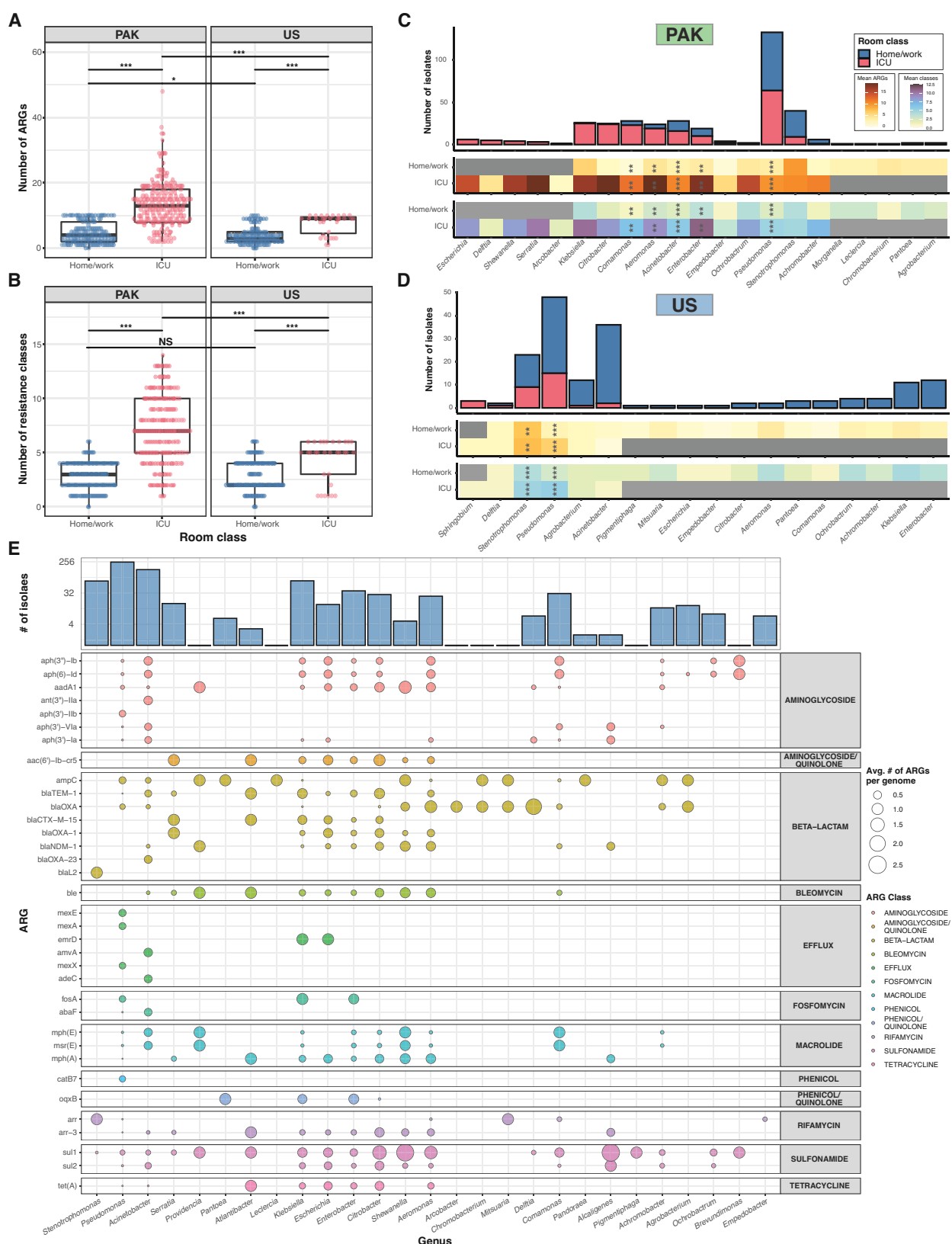

**FIG 4** Overall ARG content of genomes. (A, B) Scatterplot showing the number of ARGs and number of ARG classes per genome for all isolates (as predicted by AMRFinder), with small amount of jitter added for visibility. *** = P < 0.001, * = P < 0.05 by Wilcoxon rank-sum test. (C, D) Stacked bar plots showing absolute number of isolates collected from each genus in USA/PAK, colored by room class. Bars to the left represent genera found more in ICUs, and bars to the right

**FIG 4** (Continued)

represent genera found more in HOME/WORK rooms. Each bar is annotated with heatmaps showing the mean number of ARGs (red/orange) and ARG classes (green/blue) in each genus. *** = $P < 0.001$, ** = $P < 0.01$, * = $P < 0.05$ by pairwise Wilcoxon tests with Benjamini-Hochberg adjustment. Individual group sizes and ARG counts are noted in Supplementary Data. (E) Balloon plot showing the average number of times each ARG appears in a genome from each genus (num_ARG_appearances/num_genomes), colored by ARG class. For visibility, only the top 35 ARGs found in the data set are shown. ARG, antibiotic resistance gene; ICU, intensive care unit.

pairwise Wilcoxon test with Benjamini-Hochberg adjustment) (Fig. 4C and D). This can be partially explained by the different species found in each environment, such as the various *Pseudomonas* genomospecies that were recovered exclusively from PAK-ICU (Fig. S1), but even common species like *P. stutzeri* and *A. johnsonii* were enriched in ARGs when comparing PAK-ICU and PAK-HOME or WORK isolates (Fig. S3). This result could be explained by the long-term persistence of drug-resistant strains. To support this explanation, we observed that *P. stutzeri* Strain 3 isolates (which had <11 SNVs between them) persisted in PAK-ICUs and harbored 9–10 ARGs, including the beta-lactamases $bla_{VIM-2}$ and $bla_{OXA}$ (Fig. S4). However, *A. johnsonii* strains (for which ICU isolates were also shown to be enriched in ARGs) appeared only transiently in our sampling, even though a similar number of isolates were recovered from PAK sites. These results suggest that while persistence of dominant strains is an important mechanism for maintenance of MDROs in this environment for certain taxa, it is not the only one.

## Enterobacterales, Aeromonadales, and Alteromonadales possess a similar repertoire of ARGs

To identify ARGs that were most prevalent across different taxa (and potentially acquired horizontally), we calculated the average frequency of each ARG per genome in each genus (Fig. 4E). We observed that similar ARGs frequently appeared in the genomes of different genera. Concerningly, we found the extended-spectrum β-lactamases (ESBLs) and carbapenemases $bla_{CTX-M-15}$, $bla_{NDM-1}$, and $bla_{OXA-23}$ among the most frequent ARGs in this cohort (Fig. 4E). On average, $bla_{CTX-M-15}$ appeared $0.13\times$ to $1\times$ per Enterobacterales genome (2/15 *Escherichia* and 16/16 *Serratia* isolates, respectively). The metallo-β-lactamase (MBL) $bla_{NDM-1}$ appeared $0.8\times$ for each *Shewanella* genome (4/5), $0.69\times$ for each *Aeromonas* genome (18/26), and $0.52\times$ for every *Citrobacter* genome (15/29) (Fig. 4E). These ARGs, along with the genes *sul1*, *sul2*, *ble*, *aph(3″)-Ib*, and *aph (6)-Id* appeared to co-occur in the genomes of Enterobacterales (Fig. 4E), suggesting that HGT may play a role in persistence of ARGs in this environment.

## Clustering of shared sequences identifies widespread sharing of ARG-encoding regions

To identify potential HGT events, we developed a computational pipeline for identifying horizontally transferred sequences in our entire data set using a BLASTn-based clustering method to identify families of common plasmidic sequences (25, 35). To reduce the likelihood of hits resulting from nearly identical, vertically acquired sequences, we first selected contigs originating from plasmids using Platon on all genomes (34). We then performed an all-vs-all alignment of the selected contigs using BLAST and filtered alignments at ≥99% identity, ≥95% coverage, and ≥5 kb in length (36). We first compared the total amount of shared genomic space between isolates to taxonomy of isolates and found that genomes within the orders Enterobacterales, Aeromonadales, and Alteromonadales appear to engage in extensive cross-species sharing of plasmidic DNA (Fig. 5A). Curiously, Pseudomonadales (*Pseudomonas* and *Acinetobacter* spp.) did not appear to have as widespread sharing of DNA, despite representing a larger portion of the data set (Fig. 5A). We then annotated the ARGs encoded on shared sequences, and found that many ESBLs and carbapenemases were frequently encoded on contigs shared between different taxa (Fig. 5C). The most widespread ARG sharing occurred with contigs encoding *ble*, *sul1*, and $bla_{NDM-1}$, which were shared between >21 unique combinations of species. Moreover, $bla_{NDM-1}$ was shared between unique combinations

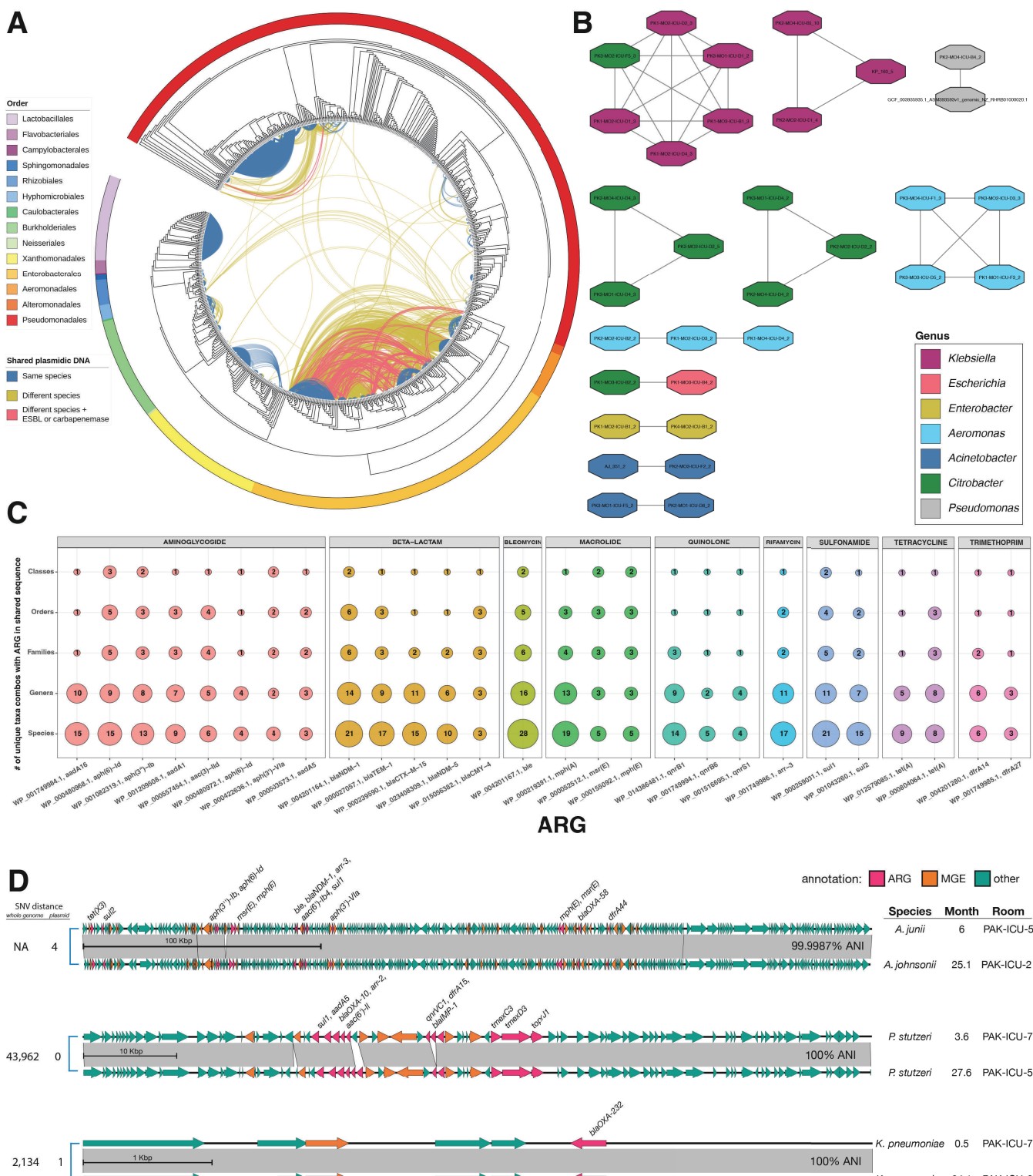

**FIG 5** HGT events within the sink environment. (A) Phylogenetic cladogram of all 822 genomes in this study, generated using GTDB-Tk and RAxML. The outer ring is colored by taxonomic order, and the lines connecting each node represent shared genomic space between those two genomes by BLAST alignment (>5 kbp in length, >99% identity, and >95% coverage). Lines are colors blue if the two genomes are the same species, yellow if they are different species, and pink if they are different species and the shared genomic spaces encodes an ESBL or carbapenemase. (B) Network showing the 11 clusters of plasmid sequences identified using nanopore sequencing and sequence alignment. Each node represents a single contig, colored by genus. An edge connecting two

**FIG 5** (Continued)

nodes represents a significant BLAST alignment between those two contigs (>5 kbp in length, >99% identity, and >95% coverage, <10% difference in contig size, at least one contig is circularized). (C) Balloon plot showing the most commonly shared ARGs different taxonomic levels. For each ARG encoded on a shared genomic space, the number of unique taxa combinations sharing that ARG was counted at different taxonomic levels. For visibility, only the most commonly-shared ARGs are shown. (D) Nucleotide alignment of plasmid Clusters 10, 7, and 2. Gray blocks show BLAST matches of >99% ID and >5 kb, with SNV counts on the left, and the ANI (based on SNVs) noted on the right. ORFs are colored by function (pink = ARG, orange = MGE, teal = other). ANI, average nucleotide identity; ARG, antibiotic resistance gene; ESBL, extended-spectrum β-lactamase; HGT, horizontal gene transfer; SNV, single-nucleotide variant.

of 14 genera, 6 families, and 6 orders (Fig. 5C). Looking closer at these results, much of this diversity was driven by sharing between *Shewanella* and various Enterobacterales. Other ARGs known to confer high levels of resistance were found on shared sequences, including $bla_{TEM-1}$, $bla_{CTX-M-15}$, $bla_{NDM-5}$, and $bla_{CMY-4}$, but these were not limited to beta-lactamases. We found sharing of nine different classes of ARGs between many taxa (Fig. 5C). These observations suggest the possibility of HGT allowing for dissemination of clinically relevant ARGs within the hospital environment.

## Plasmids harboring diverse and clinically important ARGs are disseminated across multiple genera and detected over 19 months apart

Epidemiological and genomic evidence suggested the possibility that numerous ARGs were shared among spatially linked pathogens species through plasmids. To identify such events, we used an iterative approach, where we clustered the BLAST alignment data using Cytoscape (37) and performed Oxford Nanopore Technology (ONT) long-read sequencing on representative isolates from each cluster to circularize plasmid sequences. Clusters containing multiple species and isolates annotated with ESBLs and carbapene-mases were given preference. After hybrid assembly, the all-vs-all BLAST alignment was repeated, and clusters were re-analyzed. In total, we performed nanopore sequencing on 60 isolates, which we combined with the 10 hybrid assemblies that were previously reported in D'Souza et al. (29). To identify families of shared plasmids, we further filtered the alignment results using the following criteria: (i) at least one aligned contig was circularized and (ii) the aligned contigs differed by no more than 10% in length. Overall, we identified 11 clusters (Table 1 and Fig. 5B) of highly similar plasmid families. To identify plasmid sharing within each family, we employed an SNV-based analysis used for tracking plasmid spread within healthcare-associated bacteria (35). Within each family, a reference sequence was chosen based on the first appearance of that plasmid, and Illumina reads from each sample within that family were aligned to that reference to quantify SNVs. A threshold of 99.985% (<15 SNVs per 100 kbp) was chosen to determine if the shared sequences were likely due to HGT (35).

This analysis identified a pair of nearly identical plasmids (4 SNVs over ~300 kbp) carrying the tetracycline destructase *tet* (X3), the beta-lactamases $bla_{NDM-1}$ and $bla_{OXA-58}$, and numerous aminoglycoside ARGs shared between two *A. junii* and *A. johnsonii* isolates collected in months 6 and 25, respectively (Fig. 5D). MOB-suite identified MOBP relaxases on this plasmid, and annotated it as mobilizable (Table S1), providing additional evidence that this plasmid is able to cross genetic barriers through HGT. A separate cluster of similar plasmids (Cluster 11) was also identified, which differed by a segment encoding $bla_{OXA-58}$ was found shared between two unrelated *A. johnsonii* isolates (16 plasmid SNVs vs >73,000 whole genome SNVs) from month 23 in two different rooms in PAK-ICU. This suggests that large conjugative MOBP plasmids are possibly endemic in this environment and enable ARG persistence through HGT among *Acinetobacter* spp. We also observed a pair of near-identical plasmids (0 SNVs over ~85 kb, >95% coverage) carrying the carbapenemase $bla_{IMP-1}$ shared between two unrelated *P. stutzeri* isolates (>43,000 SNVs across the entire genome) collected 24 months apart (Fig. 5D). This plasmid was annotated as "non-mobilizable" by MOB-suite, but the gene annotations revealed sequences with homology to the replication initiator protein RepA and Type IV conjugal transfer proteins (TraD, TIGR03759-family protein, TIGR03752-family protein). It is possible that sequence databases are not adequate to fully annotate the

**TABLE 1** Summary of plasmid clusters

| Cluster | Mean length (bp) | Mean GC (%) | Replicon type(s) | Relaxase type | MPF type | ARG(s) | Species in cluster |
|---|---|---|---|---|---|---|---|
| 1 | 67,328.2 | 53.3 | IncN | MOBF | MPF_T | aac(6')-Ib-cr5, aadA16, arr-3, blaNDM-5, blaTEM-1, ble, dfrA27, mph(A), qnrB6, sul1, tet(A) | K. pneumoniae, C. freundii, K. michiganensis, E. hormaechei |
| 2 | 6,141.0 | 52.2 | rep_cluster_1195 | MOBP | – | blaOXA-232 | K. pneumoniae |
| 3 | 177,388.5 | 54.4 | – | – | – | aadA1, blaNDM-1, blaOXA, blaPER-3, ble, mph(A), sul1 | A. caviae |
| 4 | 110,792.0 | 54.8 | IncFIB,IncFII,rep_cluster_2272 | MOBF | MPF_F | blaNDM-1, ble, rmtC, sul1 | C. freundii |
| 5 | 309,511.3 | 48.8 | Col(VCM04) | MOBH | – | aac(3)-IIe, aac(6')-Ib-cr5, aadA1, aadA16, arr-3, blaCTX-M-15, blaNDM-1, blaOXA-1, ble, catA2, dfrA27, sul1, sul2 | C. freundii |
| 6 | 222,581.3 | 55.1 | rep_cluster_1332 | MOBH | – | aac(3)-IId, aac(6')-Ib, aac(6')-Ib-cr5, aadA1, aadA16, ant(3")-IJ/ aac(6')-Ib, aph(3")-Ib, aph(6)-Id, arr-3, blaNDM-1, blaOXA, blaOXA-1, blaPER-3, blaTEM-1, ble, catB3, mph(A), sul1, tet(A) | A. caviae |
| 7 | 85,098.0 | 57.2 | – | – | – | aac(6')-Ib4, aac(6')-II, aadA5, arr-2, blaIMP-1, blaOXA-10, dfrA15, sul1, tmexC3, tmexD3, toprJ1 | P. stutzeri |
| 8 | 329,536.5 | 47.1 | IncHI2A,rep_cluster_1088 | MOBH | MPF_F | aac(6')-Ib-cr5, aadA1, aadA16, arr-3, blaNDM-1, blaOXA-1, ble, catA1, dfrA27, qnrB1, sul1, tet(A) | E. coli, C. freundii |
| 9 | 333,080.0 | 47.1 | IncHI2A,rep_cluster_1088 | MOBH | MPF_F | aac(3)-IIe, aac(6')-Ib-cr5, aadA1, aadA16, arr-3, blaNDM-1, blaOXA-1, blaTEM-1, ble, catA1, dfrA27, mph(A), mph(E), msr(E), qnrB1, sul1, tet(A) | E. roggenkampii, E. hormaechei |
| 10 | 331,845.0 | 38.8 | – | MOBP,MOBP | – | aac(6')-Ib4, aph(3')-Ib, aph(3')-VIa, aph(6)-Id, arr-3, blaNDM-1, blaOXA-58, ble, dfrA44, mph(E), msr(E), sul1, sul2, tet(X3) | A. junii, A. johnsonii |
| 11 | 302,786.5 | 39.3 | – | MOBP | – | aac(6')-Ib4, aph(3')-Ib, aph(3')-VIa, aph(6)-Id, arr-3, blaNDM-1, ble, mph(E), msr(E), sul1, sul2, tet(X3) | A. johnsonii |

type IV secretion system (T4SS) proteins in Pseudomonadales, or the T4SS machinery is incomplete in this plasmid and requires a helper plasmid for conjugation (38). The host isolate of this plasmid was not found to contain other plasmid sequences, however. Finally, another pair of nearly identical plasmid (one SNV over 6,141bp) carrying the $bla_{OXA-232}$ carbapenemase and a MOBP-family relaxase was found in two distantly related *K. pneumoniae* isolates (2,134 SNVs over the entire genome) collected over 23 months apart (Fig. 5D). On shorter time scales, we also observed extensive sharing of near-identical plasmid sequences (Table 1, Supplementary Data). Our analysis suggests that these plasmids are disseminated widely in this environment, can persist for up to 2 years, and can mobilize to unrelated hosts.

## Persistent plasmids confer a resistance phenotype

To investigate the phenotypic consequences of harboring some of these plasmids, we identified strains with multiple isolates that differed in the presence or absence of an identified plasmid. We performed Antibiotic Susceptibility Testing (AST) using Kirby-Bauer disk diffusion assays in accordance with Clinical and Laboratory Standards Institute (CLSI) guidelines (Supplementary Data) (39). When we tested *K. michiganensis* Strain 1 isolates, we observed that all except for one (PK1-MO2-ICU-B4) was pan-resistant to all beta-lactams tested, including the carbapenems imipenem and meropenem (Fig. S5B). WGS showed that all resistant isolates harbored the beta-lactamases $bla_{TEM-1}$ and $bla_{NDM-5}$, which PK1-MO2-ICU-B4 lacked (Fig. S5B). Nanopore sequencing of another *K. michiganensis* Strain 1 isolate collected from the same room at the same time (PK1-MO2-ICU-D4) was able to circularize the Cluster 1 plasmid harboring these ARGs (Fig. 5E Fig. S5A). This same plasmid was found in the genomes of five other Enterobacterales isolates, varying by at most one SNV: two unrelated *C. freundii*, an unrelated *K. michiganensis*, a *K. penumoniae*, and an *E. hormaechei* isolate. AST testing showed that those isolates were also not-susceptible to all beta-lactams tested, with the exception of the other *K. michiganensis* (PK3-MO2-ICU-F5) showing susceptibly to Cefotetan (Supplementary Data). This cluster of plasmids was annotated by MOB-suite as an IncN plasmid about ~67 kbp in length, and encoding a MOBF relaxase that suggests that it is a conjugative plasmid. These data provide strong evidence that presence of this ARG-harboring plasmid, which was found in numerous Enterobacterales genera and encodes conjugation machinery, is sufficient for conferring clinically relevant resistance to multiple classes of beta-lactamases.

## DISCUSSION

The colonization of hospital sink surfaces by MDROs leading to disease is well documented, but the transmission dynamics that enable these surfaces to act as reservoirs is not well understood (18, 40). Here, we present a multiyear, genomic investigation of bacterial colonization in matched clinical and non-clinical spaces in Pakistan and the USA. This resulted in the recovery of 530 new bacterial isolates, for a total of 822 isolates across both sampling sites. Using comparative genomics, we identified 104 species in total, including 37 genomospecies. Across all sites, we find a similar microbial ecology, dominated by *Pseudomonas*, *Acinetobacter*, and *Stenotrophomonas*. This is concordant with other observations (29, 41, 42), although we find a unique and diverse microbiome of taxa that are not currently represented in databases. We recovered many isolates from both clinical and non-clinical environments, but we found a pattern of increased ARG abundance and diversity in ICU sites. To our knowledge, this is the first comparative geonomic comparison of MDRO colonization in geographically matched ICU and community built environments. We identified variable colonization patterns, where some species demonstrated strain persistence on ICU sinks for over 2 years in the same building, while other species appear more transiently. Focusing on isolates that harbored beta-lactamases, we found examples of strain persistence allowing ICU sinks to act as a reservoir for ARGs. We found similar patterns of ARG content among Enterobacterales which led us to perform additional nanopore long-read sequencing on 60 isolates,

resulting in near-complete genomes and circularized plasmid sequences. By analyzing networks of highly similar plasmid sequences, we identified potential occurrences of HGT enabling ARG transfer across taxonomic boundaries and maintenance of ARGs in the environment over time. This work adds to the body of work of MDRO colonization in hospital built environments, illuminates how colonization dynamics varies between different taxa, and illuminates examples of MDRO persistence in hospital sinks both through vertical and horizontal transmission of ARGs.

Recent studies have similarly sought to understand and characterize the ecology of hospital sinks using genomics (14, 18, 42–47). These surveys have found similarities between environmental isolates and patient isolates associated with nosocomial outbreaks, particularly in *S. marcescens*, *P. aeruginosa*, and *C. freundii*. Some high-resolution studies indicate that similar strains establish themselves in sinks and persist over time, suggesting that sinks act as a reservoir for these pathogens (18, 42). Detection of similar MDROs in wastewater indicates that these pathogens can spread beyond a single sink and enter the larger ecosystem (48, 49). Based on these observations, we focused on longitudinal genomic measurements of MDROs persistence on these surfaces, in multiple rooms and buildings.

We found long-term contamination of surfaces, both from common pathogens like *P. aeruginosa* and less common, but increasingly appreciated pathogens like *P. stutzeri* (50, 51). Hospital sinks acting as a reservoir of *P. aeruginosa* has been reported before (13, 16, 46, 52, 53), but here we show that persistence of these strains is common outside of an outbreak setting, and provide strong genomic evidence showing reappearance of a recently transferred isolate rather than a derivative lineage (18). Interestingly, not all strains were found to persist over multiple collection times, and the *P. aeruginosa* accessory genome was found to be significantly associated with whether that strain was found at multiple time points (Fig. S7). We can speculate that a genomic association exists with strain persistence in *P. aeruginosa*, but elucidating that would require much more extensive sampling. We found less-extensive examples of strain persistence by Enterobacterales, which have been associated with nosocomial infections (15, 17, 54, 55). Sinks have previously been associated with nosocomial *A. junii* infection (56), and we previously found extensive colonization of PAK-ICU surfaces with *Acinetobacter* spp. (29). Similarly, we frequently recovered *A. junii* and *A. johnsonii* isolates from both clinical and non-clinical sink surfaces at PAK and US sites. So, it was somewhat surprising to find no examples of strain persistence. Both *A. junii* and *A. johnsonii* have been reported to form biofilms on liquid interfaces, which allow them to resist disinfection and persist over time (57–60). It is possible that the strains we sampled did not form biofilms as readily, or they were present at much lower density and additional sampling be necessary to detect strain persistence. Altogether, these findings are concordant with previous observations that *Pseudomonas* spp. can establish long-term colonization of sink surfaces (42, 52). Meanwhile, we frequently detected *Klebsiella* and *E. coli*, but strains appeared to be more transient in their appearance, possibly shed by recent patients or healthcare workers. Previous genomic surveillance of *Klebsiella* and *E. coli* has also shown that hospital sinks are populated by a mixture of strains, which can be associated with ward or location (61, 62), but the limited resolution of methods like MLST and core genome comparisons means that claims about vertical transmission cannot be made. Here, we observe that although sink contamination with *Klebsiella* and *E. coli* is common, the persistence of individual strains occurs on a shorter time scale compared to *Pseudomonas* (Fig. 2 Fig. S2). This corroborates observations that *Klebsiella* infection in hospitalized patients is largely caused by unrelated strains, likely originating from the patient's own microbiome (63, 64). However, nosocomial transmission is still possible and detection and disinfection of these pathogens is critical to protect susceptible patients from recently shed MDROs (55, 64).

Perhaps the most concerning finding is the maintenance of ARG-harboring plasmids in hospitals by HGT. This work adds to the growing body of literature that establishes specific examples of HGT occurring within hospitals (20, 25, 47) and describes specific

cases of HGT allowing for transfer of multiple ARGs—including carbapenemases and ESBLs—across phylogenetic boundaries (25, 47). This presents another mechanism by which ARGs persist in the environment that does not require a strain to establish itself on a surface. This is best illustrated by plasmid Clusters 2, 7, and 10 which demonstrate how high-risk carbapenemases can be preserved in species where we did not observe any strain persistence, or even between hosts of different species. When we typed these plasmid sequences, we identified relaxase and replicon types that are predicted to allow conjugation and are associated with appearance in multiple genera (65). We also identified plasmids in *Acinetobacter* spp., which are known to be naturally competent (66). Much of the sharing of DNA sequences we found was within Enterobacterales, which is consistent with the dense plasmid exchange network seen in this order (25, 26, 41, 65). We did not find HGT as frequently within Pseudomonadales despite those genomes representing almost half of our data set. Other studies have reported that shared sequences among *P. aeruginosa* and *Stenotrophomonas* spp. contain many prophages and integrated conjugative elements (41), so it is possible that our analysis (which was designed to identify shared plasmids) overlooked the mechanism of HGT that is more commonly used by these taxa.

This study had several limitations. By selective culturing to enrich for MDROs, we did not capture the full microbial ecology of these surfaces. Metagenomic studies, which are becoming increasingly common in these environments, would be better suited to comprehensively profile their respective microbiomes (18, 67). We also cannot conclude an exact mechanism of transmission, as surface-to-surface transmission would appear the same as source-to-multiple-surface in our sampling scheme. Finally, our bioinformatic analysis was designed to detect whole plasmids, and therefore discarded shared sequences that may be the result of transposition and integration in the genome. Further analysis designed to capture the full extent of HGT on these surfaces could better characterize HGT events in non-Enterobacterales isolates.

In conclusion, our investigation of MDRO persistence in hospital sinks provides a high-resolution view of the strain dynamics of differing taxa, the ARG burden in these environments, and the different mechanisms used by these taxa that results in the maintenance of ARGs in these environments. Our work illustrates the utility of genomic-based methods for monitoring surface contamination and emphasizes the role that HGT plays in the persistence of ARGs in the environment. Further work is needed to understand the full dynamics between patients, healthcare workers, and ICU sinks, complemented by efforts to decontaminate sinks and eliminate these MDRO reservoirs.

## MATERIALS AND METHODS

### Sample collection and culturing

Rooms (ICU, HOME, and WORK) were sampled every month for 5 months (due to a local holiday, the Month 4 samples from PAK were not collected). At each time point, three sink surfaces were sampled in each room: sink faucet opening (swabbing on the aerator for 1 min), sink basin (swabbing entire inside surface for 1 min), and the sink drain (swab inserted directly into the drain and rubbed against the sides of the pipe). Two water samples were then also collected: first collection (the first 14mL sitting in the fixture) and 2 min flow (after letting the water run for 2 min, collect 14mL of water). The ESwab collection and transport system (Copan, Murieta, CA, USA) was used to collect surface samples; swabs were moistened with sterile water prior to sample collection. Two swabs were held together for specimen collection. Specimens collected in Pakistan were shipped to the US site for processing and analysis. One ESwab specimen was vortexed and 90µL of eluate was inoculated to each of the following culture selective medium: VRE chromID (bioMerieux, Marcy-l'Étoile, France), HardyCHROM ESBL (Hardy Diagnostics, Santa Maria, CA, USA), Cetrimide Agar (Hardy, Santa Maria, CA, USA), MacConkey Agar with cefotaxime (Hardy, Santa Maria, CA, USA), and MacConkey Agar with ciprofloxacin (Hardy, Santa Maria, CA, USA). Sheep's blood agar (Hardy, Santa Maria, CA, USA) was also

used as a growth control, but only isolates from selective plates were sequenced. Plates were incubated at 35°C in an air incubator and incubated up to 48h prior to discard if no growth. Up to four colonies of each colony morphotype (as appropriate for the agar type) were subcultured and identified using matrix-assisted laser desorption/ionization time-of-flight mass spectrometry (MALDI-TOF MS) with the VITEK MS system. Water samples were briefly vortexed, and 100µL was inoculated on the same media. All isolates recovered were stored at −80°C in TSB with glycerol.

## Antibiotic susceptibility testing

After shipment to the US site, antimicrobial susceptibility testing was performed using Kirby-Bauer Disk Diffusion, interpreted according to criteria from the M100-S30 (39), on 329 isolates from commonly recovered.

## Illumina WGS

Total genomic DNA was obtained from pure cultures using the QIAmp BiOstic Bacteremia DNA kit (Qiagen, Germantown, MD, USA). DNA was quantified with the Quant-iT PicoGreen dsDNA assay (Thermo Fisher Scientific, Waltham, MA, USA), and 0.5ng of genomic DNA was used to create sequencing libraries with the Nextera kit (Illumina, San Diego, CA, USA) using a modified protocol (68). Samples were pooled and sequenced on the Illumina NextSeq platform to obtain 2×150 bp reads. The reads were demultiplexed by barcode and had adapters removed with Trimmomatic v0.38 (69). Processed reads were *de novo* assembled into draft genomes with Unicycler v0.4.7 (69) using default settings, and the assembly.fasta file was used for all downstream analysis. Assembly quality was verified using QUAST v4.5 (70), bbmap (71), and CheckM v1.0.13 (72). Genomes were included for analysis if the assemblies: (i) had <500 contigs (≥1,000 bp), (ii) the length of the assembly in small contigs (<1,000 bp) represented <2% of the total assembly length, and (iii) had an estimated completeness >90% and contamination <5%.

## ONT WGS

Total genomic DNA was obtained from pure cultures using the QIAmp BiOstic Bacteremia DNA kit (Qiagen, Germantown, MD, USA) with the following modifications to preserve High Molecular Weight DNA: heating step reduced to 8 min and bead-beating step reduced to 90 s. DNA was quantified with the Qubit BR dsDNA assay (Thermo Fisher Scientific, Waltham, MA, USA), and 1µg of genomic DNA was used to create sequencing libraries with the ONT Ligation Sequencing Kit (SQK-LSK109) and Native Barcodes (EXP-NBD196) (Oxford Nanopore Technologies, Oxford, UK). Samples were pooled and sequenced on an ONT MinION Flow Cell (R9.4.1 chemistry). Reads were demultiplexed and basecalled with guppy v5.0.11 using the following command: guppy_basecaller -i fast5/ -s fastq/ --config dna_r9.4.1_450bps_sup.cfg gpu_runners_per_device 24 --num_callers 12 --compress_fastq --trim_barcodes --disable_qscore_filtering --barcode_kits EXP-NBD196–-detect_mid_strand_barcodes --min_score_mid_barcodes 60x cuda:0. Long reads were filtered with filtlong v0.2.0 (73) to remove reads <1,000 bp, and the worst 5% of reads. Long reads were combined with trimmed short reads and assembled with Unicycler v0.4.7 (69) using default settings. Assemblies were subjected to the same quality control measures as before.

## Genome annotation

The assembly.fasta file from Unicycler was annotated with prokka v1.14.5 (74), and the resulting files were used for all further analyses. ARGs were identified using AMRFinder-Plus v3.9.8 (75). For statistical comparisons of ARG counts, only 530 new isolates were considered.

## Taxonomic assignment

All isolates were initially identified using the VITEK MS v2.3.3 system (bioMérieux, Marcy-l'Étoile, France). Following draft genome assembly, the species assignment was done using an *in sliico* approach. Species were first approximated by running Mash Screen (76) on each draft genome against a reference database built with all bacterial assemblies on RefSeq marked as "Assembly from type," "Assembly from synonym type," or "Reference" (accessed 16 July 2021). The top two hits from the Mash Screen results were selected as references. Those reference genomes were combined with our assembled draft genomes and compared in an all-vs-all manner using FastANI to calculate ANI between each pair of isolates (77). Species were determined if the genome in question had >95% ANI to a reference type genome (77). Isolates that did not pass this threshold were considered to be novel genomospecies, and a putative genus was assigned using the closest reference type genome matches. In addition to our assembled draft genomes, all genomes from D'Souza et al. were downloaded from RefSeq (BioProject accession PRJNA497126) (29) and included in this analysis to detect common species and genomospecies.

## Core genome alignment

For individual species, the .gff files generated by Prokka were used to construct a core genome alignment with Roary v3.12.0 and PRANK v.170427 (78, 79). The core genome alignment was used to build a maximum likelihood tree using rAxML v8.2.11 and visualized using iToL (80, 81). For the phylogenetic analysis of all isolates, marker genes were identified, extracted, and aligned using GTDB-Tk v1.7.0 (82). A maximum likelihood tree was constructed from the alignment as before using rAxML and iToL (80, 81).

## Strain-sharing analysis

For species where at least 10 isolates were found across both studies (with at least one being from this study), pairwise core genome SNP counts between isolates were calculated using snp-dists v0.8.2 and clustered roughly using a cutoff of <500 pairwise SNPs (83). Within each grouping, trimmed Illumina reads and draft genomes were used to call variants in an all-vs-all manner using snippy v4.6.0 (84). Reads from D'Souza et al. were downloaded from SRA using the BioProject accession (PRJNA497126) and processed with Trimmomatic v0.38 (69). SNVs (including SNPs and indels) were counted between each query-reference pairing, with sites showing variation between a genome and its own reads masked from all calculations. Within each grouping, the genome with the highest median number of aligned bases when used as a reference was chosen as the reference assembly for comparisons within that group. ANI was calculated using the formula:

$$\text{ANI} = \frac{(\text{num\_aligned\_bases} - \text{SNVs})}{\text{num\_aligned\_bases}}$$

An ANI cutoff of 99.9995% was chosen as a cutoff for determining strain identity based on the cutoff of 99.999% recommended by inStrain's authors (85) and the observation that it captured the comparisons between the most similar isolates in the multimodal distribution of calculate ANI values (Fig. S6). Given observations that non-hypermutator *P. aeruginosa* strains accumulate SNPs at a rate of 1.1–5.5 per year in the setting of chronic infection (86–88), this threshold would represent about 6.4–31.8 years of divergence. These strain groups were confirmed using inStrain: genomes were dereplicated with drep v3.2.2 (89) to create a reference set, reads were mapped with bowtie2 v2.3.4.1 (90), profiled with inStrain profile, and pairwise comparisons calculated with inStrain compare (19). The same threshold was used with inStrain's popANI statistic.

If there were multiple calculations for the same comparison (using a different reference), the one with the highest number of compared bases was selected.

## Accessory genome analysis of *P. aeruginosa*

Core genes were removed from the gene_presence_absence.Rtab generated by Roary. The vegdist function from the Vegan R package to calculate Jaccard distance between genomes, and pcoa function from the ape R package was used to perform principal coordinates decomposition (91). PERMANOVA was performed using the adonis2 function in Vegan (92).

## Identification of shared plasmid sequences

Draft genomes were first filtered for contigs predicted to originate from plasmids using platon v1.5 (34), and then aligned against one another in an all-vs-all manner using nucmer v4.0.0b2 (93). Matches were filtered for >5 kbp and 95% identity. Sequences were extracted and merged with bedtools v2.27.1 (94) and aligned against one another by all-vs-all blastn using blast+ v2.6.0 (95). The resulting comparisons were filtered for matches that were ≥99% identity, ≥95% coverage, and ≥5 kbp, and clustered using Cytoscape v3.9.0 (37). To identify plasmid sharing events, the comparisons were further filtered for alignments where at least one contig was circularized, and the two contigs being compared are no more than 10% different in overall length. After clusters were identified with Cytoscape, SNVs were called by aligning Illumina reads to the plasmid contig using snippy v4.6.0 within each cluster (84). The earliest appearance of the plasmid was used as reference, or the lowest percentage of unaligned bases in the case of a tie. Pairwise SNV distances were calculated based on the output of snippy-core, and ANI was calculated using the formula:

$$\text{ANI} = \frac{(\text{num\_aligned\_bases} - \text{SNVs})}{\text{num\_aligned\_bases}}$$

Pairwise plasmid alignment and visualization was done using EasyFig v2.2.0.

## Plasmid typing

Plasmid replicons, relaxases, and Mate-Pair formation types were identified using MOB-typer function in the MOB-suite v3.1.0 set of tools (96, 97).

## Annotation of MGEs on plasmids

An ORF was considered an MGE if the prokka annotation mentioned any of the following terms: "transposase," "transposon," "integrase," "integron," "conjugative," "conjugal," "recombinase," "recombination," "mobilization," "phage," "plasmid,, "resolvase," "insertion element," or "mob."

## ACKNOWLEDGMENTS

This work was supported by a United States Agency for International Development award (Award number 3220-29047) to S.A., C.-A.B.-D., and G.D. This work was also supported in part by awards to G.D. through the National Institute of Allergy and Infectious Diseases (NIAID) and the Eunice Kennedy Shriver National Institute of Child Health & Human Development of the National Institutes of Health (NIH) under Award numbers U01AI123394 and R01HD092414, respectively. L.D.-T. received support from NIAID under Award number F30AI157161. The content is solely the responsibility of the authors and does not necessarily represent the official views of the funding agencies.

The authors thank members of the Dantas lab for helpful discussions of the research and manuscript. The authors thank The Edison Family Center for Genome

Sciences & Systems Biology staff, Eric Martin, Brian Koebbe, Jessica Hoisington-López, and MariaLynn Crosby for their technical support in high-throughput computing and sequencing expertise.

## AUTHOR AFFILIATIONS

[1]The Edison Family Center for Genome Sciences and Systems Biology, Washington University School of Medicine, St. Louis, Missouri, USA

[2]Department of Pathology and Immunology, Division of Laboratory and Genomic Medicine, Washington University School of Medicine, St. Louis, Missouri, USA

[3]Atta ur Rahman School of Applied Biosciences, National University of Sciences and Technology, Islamabad, Pakistan

[4]Department of Medicine, Washington University School of Medicine in St Louis, St. Louis, Missouri, USA

[5]Department of Molecular Microbiology, Washington University School of Medicine in St Louis, St. Louis, Missouri, USA

[6]Department of Pediatrics, Washington University School of Medicine in St Louis, St. Louis, Missouri, USA

[7]Department of Biomedical Engineering, Washington University in St Louis, St. Louis, Missouri, USA

## AUTHOR ORCIDs

Luke Diorio-Toth http://orcid.org/0000-0002-6586-2811
Christopher W. Farnsworth http://orcid.org/0000-0001-7169-3850
Carey-Ann D. Burnham http://orcid.org/0000-0002-1137-840X
Gautam Dantas http://orcid.org/0000-0003-0455-8370

## FUNDING

| Funder | Grant(s) | Author(s) |
|---|---|---|
| HHS \| NIH \| National Institute of Allergy and Infectious Diseases (NIAID) | U01AI123394, F30AI157161 | Luke Diorio-Toth |
| | | Gautam Dantas |
| United States Agency for International Development (USAID) | 3220-29047 | Saadia Andleeb |
| | | Carey-Ann D. Burnham |
| | | Gautam Dantas |
| HHS \| NIH \| Eunice Kennedy Shriver National Institute of Child Health and Human Development (NICHD) | R01HD092414 | Gautam Dantas |

## DATA AVAILABILITY

Assemblies and sequencing reads associated with this report are available from NCBI under BioProject accession code PRJNA868296.

## ADDITIONAL FILES

The following material is available online.

### Supplemental Material

**Figure S1 (mSystems00206-23-s0001.eps).** Detailed breakdown of recovered isolates.
**Figure S2 (mSystems00206-23-s0002.eps).** Maximum likelihood phylogenetic tree of 7 taxa and space and time information.
**Figure S3 (mSystems00206-23-s0003.eps).** Absolute number of isolates collected from each species.

**Figure S4 (mSystems00206-23-s0004.eps).** Maximum likelihood phylogenetic tree of *P. stutzeri* isolates.

**Figure S5 (mSystems00206-23-s0005.eps).** Nucleotide alignment and AST interpretations.

**Figure S6 (mSystems00206-23-s0006.eps).** Histogram of ANI values.

**Figure S7 (mSystems00206-23-s0007.eps).** Principal coordinate analysis of *P. aeruginosa* aeruginosa colonization was not accessory genome

**Table S1 (mSystems00206-23-s0008.docx).** Individual plasmids.

**Supplemental Material Legends (mSystems00206-23-s0009.docx).** Legends for Fig. S1 to S7, Table S1, and Data Set S1.

**Data Set S1 (mSystems00206-23-s0010.xlsx).** Isolate metadata, AST results, full ARG counts.

## Open Peer Review

**PEER REVIEW HISTORY (review-history.pdf).** An accounting of the reviewer comments and feedback.

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
