## [Reviewer comments · mSystems]

Intensive care unit sinks are persistently colonized with multi-drug resistant bacteria and mobilizable, resistance-conferring plasmids

Luke Diorio-Toth, Meghan Wallace, Christopher Farnsworth, Bin Wang, Danish Gul, Jennie Kwon, Saadia Andleeb, Carey-Ann Burnham, and Gautam Dantas

Corresponding Author(s): Luke Diorio-Toth, Washington University in St Louis School of Medicine

Review Timeline:

Submission Date:	February 28, 2023
Editorial Decision:	April 14, 2023
Revision Received:	April 28, 2023
Accepted:	May 2, 2023

Editor: David Cleary

Reviewer(s): Disclosure of reviewer identity is with reference to reviewer comments included in decision letter(s). The following individuals involved in review of your submission have agreed to reveal their identity: Bede Constantinides (Reviewer #2)

Transaction Report:

DOI: <https://doi.org/10.1128/msystems.00206-23>

April 14, 2023

Dr. Luke Diorio-Toth
Washington University in St Louis School of Medicine
Edison Family Center for Genome Sciences and Systems Biology
4515 McKinley Avenue
5th Floor, Room 5121
St. Louis, MO 63110

Re: mSystems00206-23 (Intensive care unit sinks are persistently colonized with multi-drug resistant bacteria and mobilizable, resistance-conferring plasmids)

Dear Dr. Luke Diorio-Toth:

Thank you for submitting your manuscript to mSystems. We have completed our review and I am pleased to inform you that, in principle, we expect to accept it for publication in mSystems. However, acceptance will not be final until you have adequately addressed the reviewer comments.

I must ask, in particular, that you address the comment of Reviewer #2 regarding data availability.

Preparing Revision Guidelines

Please return the manuscript within 60 days; if you cannot complete the modification within this time period, please contact me. If you do not wish to modify the manuscript and prefer to submit it to another journal, please notify me of your decision immediately so that the manuscript may be formally withdrawn from consideration by mSystems.

Sincerely,

David Cleary

Editor, mSystems

Journals Department
Reviewer comments:

Reviewer #1 (Comments for the Author):

This manuscript characterizes MDROs collected from sink surfaces and water in hospital ICUs found in Pakistan and the United States. These samples were compared to those collected from sinks found in homes and workplaces located in the local community. Individual bacterial strains were isolated and assessed for antibiotic resistance profiles, including the identification of plasmids that are likely to have been shared between taxa during horizontal gene transfer events. The authors recovered more isolates from sinks in Pakistan than from the US, at a disparity that was not observed between community sites, and ARG abundance and diversity was greater on hospital sink surfaces than in home and work sites. Additionally, they were able to identify strains that were persistent in the environment over the two-year period, and they presented evidence that horizontal gene transfer events are likely to play an important role in the maintenance of ARGs in these environments over time.

Overall, I believe this to be a well-written, well-planned, and interesting manuscript. The comparison between hospitals in different countries is a unique and meaningful contribution to the literature. The lab-based methods are rigorous (e.g., the recovery of >800 isolates and the use of multiple agar types with appropriate controls), and bioinformatic methods are both sound and thoroughly considered. The conclusions are supported by their findings, and the manuscript, more generally, is presented in an organized and logical format. Figures and tables are a good addition to the text - I particularly appreciate that color palettes are color-blind friendly - and the length is appropriate for the results presented.

I have no major comments for the authors, with some minor comments below.

Minor comments:

'antibiotic resistance genes' is not hyphenated. Should either be 'antibiotic resistance genes' or 'antibiotic-resistant organisms.' Same is true of 'built environment.'

Lines 77, 79, 97, 173, etc: 'which' should be 'that' - 'that' indicates an essential part of the sentence needed to convey the meaning. 'which' is an add-on to the concept presented and should be preceded by a comma.

Line 87: Numbers 1-10 should be spelled out, so 2 should be two.

Line 92: It's not clear what '2 two' means. Two paired?

Line 107: A comma is needed after 'In this study'

Line 108: If Pakistan is presented first in the introduction, it should be listed before the US throughout.

Line 111: I would suggest removing 'We used WGS to determine that' to make the sentence flow better.

Line 118: 'commonly' to 'common'

Line 123: 'geography' should be switched to 'country', 'geographic location', or something similar

In paragraph starting at line 159: Why the switch to >24 months, when the sampling period was 25 months. Why not say 25 months, or a minimum of 25 months.

Line 225: "We also observed a pair of near-identical [plasmids]"

Line 283: 'a' needs to be un-italicized.

Line 296: "recovery [of]"

Line 317: I suggest changing "particularly in taxa like *S. marcescens*, *P. aeruginosa*, and *C. freundii*" to "particularly in *S.*

marcescens, P. aeruginosa, and C. freundii taxa" to avoid the phrase "taxa like."

Line 324: "as well as" to and

Line 336: "or [they] were present"

Line 341: Drop the 's' from infections or change the 'is' to 'are'

Line 359: Drop the 's' from contains

Line 369: 370: "could illuminate HGT [events?] in non-Enterobacterales isolates.

Line 373: 'which results' -> that result

Line 388: Were the swabs moistened with PBS?

Line 389: "workup" -> processing?

Line 403: If a sentence starts with a number, it needs to be spelled out.

Line 413: Comma needed after 'settings'

Line 417: Need an [and] before (c) and commas between each point.

Line 470: Please italicize 'et al.'

Figure 2 legend, second sentence: 'Collected' means timepoint?

Supplementary figure 7: "strains the were recovered" needs to be fixed.

Reviewer #2 (Comments for the Author):

The authors present a detailed analysis of a culture-based prospective study of sink drain colonisation patterns from intensive care units, private homes and office buildings in the United States and Pakistan. The analysis presented is of high quality and will be of broad interest to journal's readership. I commend the author's application of suitable state of the art methods and especially their presentation of high quality figures which greatly enhance the manuscript. While it is unfortunate that metagenomics was not employed in this analysis, this is nevertheless an interesting and high quality piece of work.

I wish to draw attention to two concerns:

Firstly, the authors state that the sequence data is publicly available, however the accession PRJNA868296 contains no publicly available sequences. Submitting papers for review containing false statements about data availability is unacceptable, and I cannot recommend acceptance of this article until the both assemblies and sequence data are deposited and available as has been claimed.

Secondly (minor): in light of the relatively small body of existing work about hospital sink drain colonisation with MDROs, perhaps more of the existing works with similar conclusions to this manuscript could have been cited.

Typos:

Line 448: "refereces"

This manuscript characterizes MDROs collected from sink surfaces and water in hospital ICUs found in Pakistan and the United States. These samples were compared to those collected from sinks found in homes and workplaces located in the local community. Individual bacterial strains were isolated and assessed for antibiotic resistance profiles, including the identification of plasmids that are likely to have been shared between taxa during horizontal gene transfer events. The authors recovered more isolates from sinks in Pakistan than from the US, at a disparity that was not observed between community sites, and ARG abundance and diversity was greater on hospital sink surfaces than in home and work sites. Additionally, they were able to identify strains that were persistent in the environment over the two-year period, and they presented evidence that horizontal gene transfer events are likely to play an important role in the maintenance of ARGs in these environments over time.

Overall, I believe this to be a well-written, well-planned, and interesting manuscript. The comparison between hospitals in different countries is a unique and meaningful contribution to the literature. The lab-based methods are rigorous (e.g., the recovery of >800 isolates and the use of multiple agar types with appropriate controls), and bioinformatic methods are both sound and thoroughly considered. The conclusions are supported by their findings, and the manuscript, more generally, is presented in an organized and logical format. Figures and tables are a good addition to the text – I particularly appreciate that color palettes are color-blind friendly – and the length is appropriate for the results presented.

I have no major comments for the authors, with some minor comments below.

Minor comments:

‘antibiotic resistance genes’ is not hyphenated. Should either be ‘antibiotic resistance genes’ or ‘antibiotic-resistant organisms.’ Same is true of ‘built environment.’

Lines 77, 79, 97, 173, etc: ‘which’ should be ‘that’ – ‘that’ indicates an essential part of the sentence needed to convey the meaning. ‘which’ is an add-on to the concept presented and should be preceded by a comma.

Line 87: Numbers 1-10 should be spelled out, so 2 should be two.

Line 92: It’s not clear what ‘2 two’ means. Two paired?

Line 107: A comma is needed after ‘In this study’

Line 108: If Pakistan is presented first in the introduction, it should be listed before the US throughout.

Line 111: I would suggest removing ‘We used WGS to determine that’ to make the sentence flow better.

Line 118: ‘commonly’ to ‘common’

Line 123: ‘geography’ should be switched to ‘country’, ‘geographic location’, or something similar

In paragraph starting at line 159: Why the switch to >24 months, when the sampling period was 25 months. Why not say 25 months, or a minimum of 25 months.

Line 225: “We also observed a pair of near-identical [plasmids]”

Line 283: ‘a’ needs to be un-italicized.

Line 296: “recovery [of]”

Line 317: I suggest changing “particularly in taxa like *S. marcescens*, *P. aeruginosa*, and *C. freundii*” to “particularly in *S. marcescens*, *P. aeruginosa*, and *C. freundii* taxa” to avoid the phrase “taxa like.”

Line 324: “as well as” to and

Line 336: “or [they] were present”

Line 341: Drop the ‘s’ from infections or change the ‘is’ to ‘are’

Line 359: Drop the ‘s’ from contains

Line 369: 370: “could illuminate HGT [events?] in non-Enterobacterales isolates.

Line 373: ‘which results’ -> that result

Line 388: Were the swabs moistened with PBS?

Line 389: “workup” -> processing?

Line 403: If a sentence starts with a number, it needs to be spelled out.

Line 413: Comma needed after ‘settings’

Line 417: Need an [and] before (c) and commas between each point.

Line 470: Please italicize ‘et al.’

Figure 2 legend, second sentence: ‘Collected’ means timepoint?

Supplementary figure 7: “strains the were recovered” needs to be fixed.

Response to Reviewer Comments

Diorio-Toth *et al.*

Reviewer #1 (Comments for the Author):

This manuscript characterizes MDROs collected from sink surfaces and water in hospital ICUs found in Pakistan and the United States. These samples were compared to those collected from sinks found in homes and workplaces located in the local community. Individual bacterial strains were isolated and assessed for antibiotic resistance profiles, including the identification of plasmids that are likely to have been shared between taxa during horizontal gene transfer events. The authors recovered more isolates from sinks in Pakistan than from the US, at a disparity that was not observed between community sites, and ARG abundance and diversity was greater on hospital sink surfaces than in home and work sites. Additionally, they were able to identify strains that were persistent in the environment over the two-year period, and they presented evidence that horizontal gene transfer events are likely to play an important role in the maintenance of ARGs in these environments over time.

Overall, I believe this to be a well-written, well-planned, and interesting manuscript. The comparison between hospitals in different countries is a unique and meaningful contribution to the literature. The lab-based methods are rigorous (e.g., the recovery of >800 isolates and the use of multiple agar types with appropriate controls), and bioinformatic methods are both sound and thoroughly considered. The conclusions are supported by their findings, and the manuscript, more generally, is presented in an organized and logical format. Figures and tables are a good addition to the text - I particularly appreciate that color palettes are color-blind friendly - and the length is appropriate for the results presented.

We thank the reviewer for their positive assessment of our work!

I have no major comments for the authors, with some minor comments below.

Minor comments:

'antibiotic resistance genes' is not hyphenated. Should either be 'antibiotic resistance genes' or 'antibiotic-resistant organisms.' Same is true of 'built environment.'

Fixed

Lines 77, 79, 97, 173, etc: 'which' should be 'that' - 'that' indicates an essential part of the sentence needed to convey the meaning. 'which' is an add-on to the concept presented and should be preceded by a comma.

Fixed

Line 87: Numbers 1-10 should be spelled out, so 2 should be two.

Fixed

Line 92: It's not clear what '2 two' means. Two paired?

Fixed, that was a typo.

Line 107: A comma is needed after 'In this study'

Fixed

Line 108: If Pakistan is presented first in the introduction, it should be listed before the US throughout.

Fixed

Line 111: I would suggest removing 'We used WGS to determine that' to make the sentence flow better.

Done

Line 118: 'commonly' to 'common'

Fixed

Line 123: 'geography' should be switched to 'country', 'geographic location', or something similar

Done

In paragraph starting at line 159: Why the switch to >24 months, when the sampling period was 25 months. Why not say 25 months, or a minimum of 25 months.

The sampling period between the two studies was about 27 months and did not occur in exact one-month intervals, so we tried to simplify this using a common minimum round number. Rounded to the nearest whole month and added "about". The exact collection dates are provided as supplementary data.

Line 225: "We also observed a pair of near-identical [plasmids]"

Assuming this comment is referring to line 255, added missing word.

Line 283: 'a' needs to be un-italicized.

Fixed

Line 296: "recovery [of]"

Fixed

Line 317: I suggest changing "particularly in taxa like *S. marcescens*, *P. aeruginosa*, and *C. freundii*" to "particularly in *S. marcescens*, *P. aeruginosa*, and *C. freundii* taxa" to avoid the phrase "taxa like."

Done

Line 324: "as well as" to and

Done

Line 336: "or [they] were present"

Fixed

Line 341: Drop the 's' from infections or change the 'is' to 'are'

Fixed

Line 359: Drop the 's' from contains

Fixed

Line 369: 370: "could illuminate HGT [events?] in non-Enterobacterales isolates.

Fixed

Line 373: 'which results' -> that result

Fixed

Line 388: Were the swabs moistened with PBS?

The swabs were moistened with sterile water before sampling, added clarification on Line 394. The “eluate” referred to in Line 396 refers to the preservation liquid shipped with the ESwab system. After sampling, each swab was placed in a screw cap tube prefilled with liquid Amies. The swab stick was broken in half, retaining the sampling end in the tube, and then sealed and sent to the US site for processing.

Line 389: "workup" -> processing?

Done

Line 403: If a sentence starts with a number, it needs to be spelled out.

Fixed

Line 413: Comma needed after 'settings'

Fixed

Line 417: Need an [and] before (c) and commas between each point.

Fixed

Line 470: Please italicize 'et al.'

Fixed

Figure 2 legend, second sentence: 'Collected' means timepoint?

It was meant to say “collection”, however, timepoint is clearer so we changed to that.

Supplementary figure 7: "strains the were recovered" needs to be fixed.

Fixed

Reviewer #2 (Comments for the Author):

The authors present a detailed analysis of a culture-based prospective study of sink drain colonisation patterns from intensive care units, private homes and office buildings in the United States and Pakistan. The analysis presented is of high quality and will be of broad interest to journal's readership. I commend the author's application of suitable state of the art methods

and especially their presentation of high quality figures which greatly enhance the manuscript. While it is unfortunate that metagenomics was not employed in this analysis, this is nevertheless an interesting and high quality piece of work.

I wish to draw attention to two concerns:

Firstly, the authors state that the sequence data is publicly available, however the accession PRJNA868296 contains no publicly available sequences. Submitting papers for review containing false statements about data availability is unacceptable, and I cannot recommend acceptance of this article until the both assemblies and sequence data are deposited and available as has been claimed.

We sincerely apologize for this, we did not intend to mislead readers. The referenced BioProject accession listed was registered and we intended to time the public release of the data with publication of the manuscript. All assemblies are uploaded to GenBank/RefSeq and the raw sequencing reads are uploaded to SRA. We have requested NCBI release all data, however it may take several weeks for them to process everything. Please do not hesitate to contact us if the data remains unavailable.

Secondly (minor): in light of the relatively small body of existing work about hospital sink drain colonisation with MDROs, perhaps more of the existing works with similar conclusions to this manuscript could have been cited.

Added additional citations and context to Discussion (Lines 324-327, 342-351).

Typos:

Line 448: "refereces"

Fixed

May 2, 2023

Dr. Luke Diorio-Toth
Washington University in St Louis School of Medicine
Edison Family Center for Genome Sciences and Systems Biology
4515 McKinley Avenue
5th Floor, Room 5121
St. Louis, MO 63110

Re: mSystems00206-23R1 (Intensive care unit sinks are persistently colonized with multi-drug resistant bacteria and mobilizable, resistance-conferring plasmids)

Dear Dr. Luke Diorio-Toth:

Thank you for submitting the revisions and I am pleased to tell you that your manuscript has been accepted, and I am forwarding it to the ASM Journals Department for publication. For your reference, ASM Journals' address is given below. Before it can be scheduled for publication, your manuscript will be checked by the mSystems production staff to make sure that all elements meet the technical requirements for publication. They will contact you if anything needs to be revised before copyediting and production can begin. Otherwise, you will be notified when your proofs are ready to be viewed.

If you would like to submit a potential Featured Image, please email a file and a short legend to msystems@asmusa.org. Please note that we can only consider images that (i) the authors created or own and (ii) have not been previously published. By submitting, you agree that the image can be used under the same terms as the published article. File requirements: square dimensions (4" x 4"), 300 dpi resolution, RGB colorspace, TIF file format.

We recognize that the video files can become quite large, and so to avoid quality loss ASM suggests sending the video file via <https://www.wetransfer.com/>. When you have a final version of the video and the still ready to share, please send it to mSystems staff at msystems@asmusa.org.

Sincerely,

David Cleary
Editor, mSystems

Journals Department
E-mail: mSystems@asmusa.org